# Evaluation of the Use of the 12 Bands vs. NDVI from Sentinel-2 Images for Crop Identification

**DOI:** 10.3390/s23167132

**Published:** 2023-08-11

**Authors:** Adolfo Lozano-Tello, Guillermo Siesto, Marcos Fernández-Sellers, Andres Caballero-Mancera

**Affiliations:** Quercus Software Engineering Group, Universidad de Extremadura, 10003 Cáceres, Spain; guillermosiesto@unex.es (G.S.); marcosfs@unex.es (M.F.-S.); andrescm@unex.es (A.C.-M.)

**Keywords:** remote sensing, crop classification, machine learning, deep neural networks, Sentinel-2, satellite imagery, multispectral, multitemporal, NDVI, common agricultural policy

## Abstract

Today, machine learning applied to remote sensing data is used for crop detection. This makes it possible to not only monitor crops but also to detect pests, a lack of irrigation, or other problems. For systems that require high accuracy in crop identification, a large amount of data is required to generate reliable models. The more plots of and data on crop evolution used over time, the more reliable the models. Here, a study has been carried out to analyse neural network models trained with the Sentinel satellite’s 12 bands, compared to models that only use the NDVI, in order to choose the most suitable model in terms of the amount of storage, calculation time, accuracy, and precision. This study achieved a training time gain of 59.35% for NDVI models compared with 12-band models; however, models based on 12-band values are 1.96% more accurate than those trained with the NDVI alone when it comes to making predictions. The findings of this study could be of great interest to administrations, businesses, land managers, and researchers who use satellite image data mining techniques and wish to design an efficient system, particularly one with limited storage capacity and response times.

## 1. Introduction

In many regions of industrialised countries (for example, the region of this study: Extremadura, Spain), the number of farmers and farms is in decline (https://www.ine.es/dynInfo/Infografia/Territoriales, accessed on 3 July 2023), despite the great importance of agro-livestock farming. Many local governments are promoting different technologies to improve and facilitate the work of farmers.

For several years, the application of remote sensing in agriculture has been a strategic objective for many governments, aimed at improving the production of and controlling agricultural areas. In 2014, the European Commission’s Directorate-General for Agriculture and Rural Development introduced new regulations [1] in the field of automated land control, particularly in agriculture. They proposed a procedure for the regular and systematic observation, tracking, and evaluation of all eligibility criteria, commitments, and other obligations. This monitoring can be applied over time, supported by Copernicus Sentinel satellite data or equivalent, and allows for conclusions to be drawn regarding the eligibility of the aid or support requested as needed. These crop identification processes from satellite images have been utilised for years by administrations, companies, and researchers, not only to identify crops in desired areas but also to detect illegal plantations, control the correct growth of crops, and help to improve production. With the use of data mining techniques applied to these images, remarkable results have been achieved, especially when multiple crop areas and time series data are used. In these cases, crop identification systems must be designed to be as efficient as possible, especially if there are restrictions on storage capacity and response times to carry out the process. 

When designing crop identification systems for large areas and a wide variety of crops, it is essential to choose the right technology and data with which to achieve the project objectives. It is necessary to evaluate the degree of precision and success in the identification balanced with the time required to carry out the process and the storage resources used. One of the aspects to consider is whether it is necessary to use all of the information from the satellite bands or whether it is enough to use a significant index that is widely used in crop identification, such as the normalised difference vegetation index (NDVI). In this study, we compare the results of the identification of crops using the reflectance values of the 12 bands of the Sentinel satellite with the results of only the NDVI in order to help system designers make a decision about which approach to take. The NDVI is one of the most widely used indices in remote sensing, as shown in the works referenced below.

There are several state-of-the-art classification systems, many of which, like those in [2,3], are based on classifications derived from a single image without clouds. Some studies work directly with RGB images for land cover classification [4]. Others, such as that in [5], make use of multiple images to create an NDVI time series, which is used for further classification with random forest, using 21 images from the Landsat-8 satellite. Comparable studies, like [6], implemented similar approaches with time series in China, using different sensor datasets tested with random forest and support vector machines. The use of the NDVI in time series for crop detections is endorsed in numerous studies, such as [7], even using diverse spatial resolutions, ranging from 3 m [8] to more than 30 m [9]. In addition, ref. [10] shows that crop classification using multitemporal images with the combination of the 12 bands of the Sentinel-2 satellite provides better results than using individual bands or a single image. Previous studies have shown the good performance of using all 12 bands in a time series [11]. Even articles like [12] recommend the use of all 13 bands instead of individual bands for better performance. Studies that add time series phenology features in order to improve performance, such as [13,14], show that using these data provides better accuracy, albeit marginally, than using red, near-infrared spectroscopy (NIR), and short-wave infrared (SWIR) bands. Using all of these bands already provides much higher values than the individual use of the NDVI or the normalised difference water index (NDWI). The NDVI has also been compared and combined with different statistical detection systems, such as principal component analysis (PCA) [15].

Taking all of these investigations into account, it becomes clear how important the selection of data inputs for crop identification is, depending on the project objectives. As expected, from a large number of training examples the models of each crop are obtained with high success rates. Furthermore, it is expected that the models created with information from all of the bands will be better than those from an index generated through the combination of only two. However, no studies have been found that make these comparisons, and therefore the following question arises: is it worth using the 12 bands in crop models or is it enough to use models generated only with the NDVI? This was the motivation to undertake this study—that it may serve other projects that have the same approach or other researchers who carry out similar work.

In this paper, we present a comparison between the use of the 12 bands of the Sentinel-2 satellite (that is, all of the information available) and the use of the NDVI as the only data input for the learning process, with the information as a temporal series. The results will help researchers to decide the most appropriate way to approach the crop identification process according to the requirements of their projects.

In the present paper, Section 2 outlines the proposed method, including the working area, data collection and selection, data processing, and neural network design. Section 3 encompasses the results and discussion, featuring the results obtained from the training of the generated models, a comparison of the results from the NDVI and 12-band models, and a discussion of the findings. The study ends with Section 4, which summarises the contributions and results of the study.

## 2. Method

The method followed to carry out this study is based on the INTELIPAC project, described in [16]. INTELIPAC serves to corroborate the crops declared by farmers to proceed with the payment provided for in the Common Agricultural Policy (CAP). For this reason, each neural network model is generated for each crop (in this study, we selected only 10 crops), and the output of each network is the predicted value of each pixel of a plot to the corresponding declared crop model. The main objective of this study was to evaluate the difference in the efficiency of models with 12-band data versus only the NDVI, and the advantages as well as disadvantages of both approaches. It should be noted that this study was carried out using neural networks with a simple configuration (described later, in Section 2.6), although it could have been carried out with other techniques.

The processes that are carried out for crop identification from Sentinel-2 images, for both types of inputs, are described in this section. There are two basic types of image analysis regarding crop monitoring: pixel-based and object-based [14]. For this study, a pixel-based approach was used, where each individual pixel of the plot is processed and its crop is identified. In this way, the proposed method can detect if a part of the selected area has irrigation problems, diseases, or is illegally planted with another crop, among other factors. The methodology employed consists of inputting all of the pixel data of the selected crop over a time period with their categories into the neural networks.

The proposed method is versatile and can be adapted to the specific requirements of the detection project. Some projects may need greater precision, and others may have to handle large amounts of data with a limited response time. On the other hand, others may want to focus on specific time periods with the aim of providing the early detection of the crop, analysis of activity in enclosures, or specific stages of crop evolution, such as ploughing, growth, or harvesting.

Therefore, one of the key factors for adapting the method to the project is to determine the optimal period of analysis of the images, and to weigh the precision against the execution time. The approach can range from processing a large number of images from the entire phenological cycle of a crop to opting for single-temporal data processing.

One of the phases included in the method is the search for the optimal analysis period. In [17], this process is described, and crop identification comparison data are offered using data from different periods, where the analysis period is gradually modified through heuristic searches.

This paper presents the results of two experiments. One of them consisted of the study of the 12 bands extracted from Sentinel-2, and the other of the calculation and analysis of the NDVI. The overview of the processes carried out in the experiments can be seen in Figure 1. The first procedure consisted of inputting the 12 Sentinel-2 frequency bands over a period of time, and the second one of only inputting the NDVI over the same period. All of these image data, prior to processing, were downloaded using the Sentinel-2 API, then trimmed into smaller images to avoid RAM overloads. After creating a time series with the information for each of the days, the NDVI was calculated (since Sentinel-2 images do not have this indicator precalculated). This calculation process is always carried out because we use the NDVI to display a time series graph showing the evolution of the NDVI over the months for each crop. Between these procedures there was also a treatment of anomalies, such as cloud masks, as well as normalisation and interpolation to reduce possible noise. Finally, the bands (or NDVI) were extracted and separated into training, validation, and testing data, in order to create inputs for the network. These processes have been developed and explained in previous studies and can be seen in detail in [16]. The procedures briefly stated above will be explained in subsequent subsections.

For each of these experiments, the neural network was modified based on trial and error. To achieve the best result according to the input, there is a process of tuning hyperparameters and layers based on the input dimensions. As the two kinds of inputs are so different, it would be unfair to use the same architecture for the different inputs.

A short description of the final network for each of the cases can be seen in Section 2.7. Once the training was finished, the test data were passed over the neural networks. The final results and discussions are shown in Section 3.

### 2.1. Work Area: Region of Extremadura, Spain

The experiment was carried out in Badajoz, one of the two provinces of Extremadura, the largest province in Spain, with an area of 21,766 km^2^, at degrees/minutes/seconds (DMS) coordinates of 38°40′0″ N, 6°10′0″ W (Figure 2). This region has an average temperature of 17.9 °C, with an annual average rainfall of 456 mm and an average humidity of 59.08% (https://es.climate-data.org/europe/espana/extremadura/badajoz-714920/, accessed on 3 July 2023). The highest elevation that this study included is 1112 m above sea level (MASL), and the lowest is 150 MASL.

### 2.2. Data Collection

The location and characteristics of the plots of interest were provided by the Regional Government of Extremadura (Figure 3), where each plot shows its crop label. A total of 549,579 plots were studied. In this way, it was possible to organise the information to create different classes in order to train the neural networks. All of the images corresponding to these plots had to be selected and downloaded from 1 October 2020 to 15 July 2021, and were provided by Sentinel-2 satellites.

The API (https://scihub.copernicus.eu/, accessed on 3 July 2023) provided by the ESA service was used to obtain all of these images. Our proposed method requires several images to process the temporal evolution of each pixel. If the original Sentinel images were used, there would be RAM oversaturation and internal storage problems; therefore, memory optimisation became a priority. Once all of the images had been downloaded, with a total of 1483 original images, they were processed by trimming them into more manageable areas to avoid RAM overflows, transforming the 110 × 110 km size into slices of 5 × 5 km (5000 × 5000 m), as represented in Figure 2b. For this slicing process the same coordinate system as the Sentinel satellites (SRC) EPSG:32630, World Geodetic System 1984/Universal Transverse Mercator (WGS 84/UTM) zones 30N and 29N, was used. As each of the zones covered 25 square kilometres and there were 977 zones, the total number of hectares to be processed was approximately 2,442,500 ha, but the total number of hectares corresponding to crops, and therefore actually used to feed the network, was 1,227,691.87 ha.

### 2.3. Preprocessing

After collecting the images, the data needed to be extracted from the bands. All bands were resized to a resolution of 10 m, and additionally normalised from 0 to 1 in case there was any anomaly in the original data. According to ESA documentation (https://step.esa.int/thirdparties/sen2cor/2.5.5/docs/S2-PDGS-MPC-L2A-IODD-V2.5.5.pdf, accessed on 11 July 2023), reflectance values are between 0 and 1, but a manual study of the original images, with the support of Sentinel Hub documentation (https://docs.sentinel-hub.com/api/latest/data/sentinel-2-l2a/, accessed on 11 July 2023), showed that the reflectance values can be above 1, so the abovementioned normalisation should be performed. One of these bands, band number 10 (B10), was lost because it was used for atmospheric correction, as shown in the ESA documentation.

Once the data collection and storage process of the 12 satellite bands had been carried out, the massive calculation of the NDVI of each pixel was performed and was also stored in the database. The calculation of the NDVI was created using a well-known formula (Formula (1)), where NIR (B8 band) is near-infrared reflectance and RED (B4 band) is visible red reflectance. The total time required to calculate all of the NDVIs for this experiment over a twelve-month period was about 102 h. This process was carried out in local mode, without using external software tools, to compare the times without depending on other factors, such as the times of sending and receiving images to external APIs.

Formula (1). Formulate the calculation to obtain the NDVI:(1)$$\mathrm{NDVI}=\frac{NIR-RED}{NIR+RED}$$

To appreciate the evolution of the values of the 12 bands and the NDVI, two graphs were generated (Figure 4a,b), averaging the reflectance values for each pixel of the images over the temporal period of the corn crop during 2021 on different dates, where the *y*-axis shows the spectral reflectance and the *x*-axis shows the different dates. Some anomalous data (like those seen around the date of 1/21) do not follow the trend of the curve due to the existence of atmospheric phenomena, such as clouds, which make it impossible to obtain real terrain information.

### 2.4. Crop Survey and Categories

For the experimentation process to be sufficiently complete, ten crops (corn, rice, rapeseed, chickpea, alfalfa, tomato, onion/garlic/leek, olive, vineyard, and vineyard-olive) in the province of Badajoz, Spain, were selected for the study, each with a different number of pixels (Figure 5). The graphical representation of the plot and the crop labels for each plot (commonly referred to as ground truth) were provided by the Regional Government of Extremadura, Spain, and were used for the training and evaluation process. For each given plot, the internal pixels of the polygon were calculated for further training and prediction. We had a total of 1,790,716 hectares.

Not all crops have the same sowing, growing, ripening, and harvesting periods. Internally, the crops were divided into periods (Figure 6), which correspond to the most important and representative months for crop detection and allow greater flexibility when training models with different crops. The suitable sensing time period for the identification of these crops was chosen using expert agronomists’ criteria. Two periods were created: one period, called period 1, with a start date of 1 October 2020 and an end date of 15 July 2021, and a second period, called period 2, with a start date of 1 April 2021 and an end date of 15 July 2021.

### 2.5. Training and Testing Sets

The dataset used for the experiment was created from scratch. All of the pixels were extracted from the plots for each of the ten crops used for training (corn, rice, rapeseed, chickpea, alfalfa, tomato, onion/garlic/leek, olive, vineyard, and vineyard-olive). The varieties of each of the crops were unified and treated as one, whose phenological phases include periods between 1 October 2020 or 1 April 2021 (depending on the crop) and 15 July 2021. It is worth noting that all of the input data were processed at the pixel level.

Each of the crops had their own model. The positive examples were those of the crop itself, and the negative examples were a homogeneous distribution of each of the other different crops. Pixels not belonging to any of the ten crops listed were ignored for the training in this study. Once all of the necessary data were selected, collected, and processed, they were split into 70% for training, 15% for validation, and the remaining 15% for testing (Table 1). To preserve randomness by ensuring the replication of the experiment, the data were randomly split using a fixed seed with the Keras library (https://keras.io/api/, accessed on 3 July 2023). Both majority and minority crops were included, having different amounts of data, the one with the most training pixels (7,986,370) being vineyard and the one with the fewest training pixels (54,110) being onion/garlic/leek.

### 2.6. Data Processing

Data from each of the spectrum bands collected by the Sentinel-2 satellites and of the calculated NDVI for each of the days studied comprised the inputs. The reference period was 5 days. In cases where there were two images between those 5 days (because there can be images every 2 and 3 days in some areas), an average of the two images was made. In cases where there were no images during the 5 days due to clouds or atmospheric conditions, interpolation was used, although this was not common in the work area, so the interpolation impact is not significant. A more detailed explanation of these processes can be seen in [11]. The data for the NDVI were previously calculated, inserted into the database, linked to the image from which those data had been extracted, and processed.

First and foremost, as stated in Section 2.2, the obtained original image data at the L2A level were cropped and preprocessed—as explained in [16]—discarding all of the pixels that contained clouds or erroneous data to avoid garbage in the training, validation, or testing data. These discards were made based on the scene classification mask provided by the Sentinel images. In addition, a Hampel filter [18] was applied as a data quality check, detecting remaining anomalous data (outliers) and replacing them.

Then, before training the model, an interpolation was made of all the days to avoid very sudden changes caused by data loss which could affect the neural network. Although there are different interpolation techniques, such as those in [19,20], a linear interpolation was applied. The interpolation process guarantees a complete temporal series for all of the data and prevents the training from being compromised. Moreover, the days indicating the start and end of the input are given by the periods, which indicate the expert-selected dates that best distinguish the crop. The graphical image shown in Figure 7a depicts the process of obtaining and processing information in order to convert it into an input. After downloading all of the images for the corresponding dates, the same pixel was analysed over time for each of the bands, resulting in the image to be used as an input. In the case of the NDVI (Figure 7b), the same process was used, but instead of for each band, only for the information of the new previously processed and created NDVI band.

In order to obtain an overview of the individual inputs according to crop, an average of each of the input data was calculated for each of the crops. The result for the 12 bands can be seen in Figure 8, and that for the NDVI in Figure 9.

### 2.7. Network Design

Different neural network models were designed for both the 12-band and NDVI approaches because the input dimensions are so different that training with the same model would have been ineffective. A model was trained for each of the crops; in other words, there are as many models as crops to identify. The input to each model consists of positive examples of the target crop and negative examples of each of the remaining crops. The output of each of the models consists of one predicted value representing the similarity percentage with the evaluated crop.

Each of the architectures was searched through trial and error to configure the ideal hyperparameters using a grid search hyperparameter methodology, which consisted of a search for parameters by incrementing and decrementing them in a uniform and sequential manner. Several tests were performed for each of the models until reaching the desired architecture. The results of the final architecture were as follows: 12-band models consisting of a three-layer architecture with 128, 64, and 32 layers (Figure 10a), and NDVI models also consisting of three layers but with 32, 16, and 8 layers (Figure 10b). Between each of their layers, batch normalisation and dropout to avoid overfitting were applied; the optimiser chosen was Adam [21], and, for each layer, the activation function was ReLu [22]. On the more technical side, the technology used for the development of the networks was the Keras library in Python 3, which uses TensorFlow at a low level.

Once the networks had been built and trained, different models were generated, as seen in Section 3. Based on these models, various results were obtained to be used by the administration. Different metrics were utilised in the evaluation process, such as precision (the quality of a positive prediction made by the model), recall (fraction which the model correctly identifies as true positives), accuracy (fraction of predictions the model was correct about), F-score [23] (evaluation metric that measures a model’s accuracy, which combines the precision and recall values), and loss [24] (penalty for a bad prediction).

For several years we have been studying the progress of the different crops [16], according to the statement from the CAP agency of the Regional Government of Extremadura, which includes ground truth data from inspectors. We have evaluated the accuracy of the results with these data.

## 3. Results and Discussion

This section shows the data from applying the models of the ten crops (corn, rice, rapeseed, chickpea, alfalfa, tomato, onion/garlic/leek, olive, vineyard, and vineyard-olive), using both the 12 bands of Sentinel-2 and the calculated NDVI. The results of the time, efficiency, and precision of the application of neural networks are shown with comparative graphs so that the differences can be appreciated.

### 3.1. Results of Applying the Models to 12-Band versus NDVI Inputs

The first column of Table 2 presents the training times for each of the crops for the 12-band models, with the longest time for the 12 bands being 26 min 48 s and the shortest time being 7 min 33 s, with an average time of 12 min 23 s.

The data obtained for each of the crops as a result of training the models with 12 bands are shown in Table 3. The accuracy, precision, recall, and F-score, the higher their number the better their values, show results higher than 0.89. On the other hand, we have the loss, where smaller is better. In this case, the values are less than 0.37. The mean values can be seen in the last row of Table 3.

Confusion matrices were also generated (Figure 11), showing the highest value of true positives as being 0.98 and the lowest as being 0.92. In the same way, we found the highest value of 0.98 and the lowest of 0.83 to be true negatives. In each of the confusion matrices, both the negative and positive responses include adverse crop samples.

The second column of Table 2 shows the training times for each of the crops for the NDVI models, with the longest time being 16 min 42 s and the shortest 59 s, with an average time of 5 min 2 s.

The data obtained for each of the crops as a result of training the NDVI models are shown in Table 4. Accuracy, precision, and F-score show results higher than 0.85. Loss values were less than 0.36. The mean values can be seen in the last row of Table 4.

The matrices were also generated with these tests (Figure 12), with the highest value of true positives being 0.95 and the lowest being 0.86. As for true negatives, we found the highest value as being 0.99 and the lowest as being 0.83.

### 3.2. Discussion

There are several studies that carry out analyses using the time series of various indices, such as [25], whereas we highlight those that focus exclusively on the NDVI [26,27] because it is one of the most used indices. Other studies work directly with the spectral bands extracted from the images, as can be seen in [28,29]. These studies cannot be compared with each other since, by not applying models with the same data, the comparison would be unfair.

Other investigations propose different studies comparing indices, or even comparing indices and bands, but for other purposes. The proposal of [30] performs a comparison of different indices to monitor soil salinity. The comparison presented by [31] does relate to spectral bands and indices, but focuses on quantifying the soil organic matter content. It is important to emphasise that these comparisons are not applied with the aim of improving crop identification.

Likewise, no studies similar to the one proposed here have been found to be able to carry out an exhaustive comparison that can serve systems and researchers to make a decision about the use of one of the approaches, taking into account the fact that the objective is not to establish the best detection system but to compare different types of inputs for the models.

Once all of the data necessary for the study had been calculated, it was possible to proceed with the data comparison. Looking at the results, the two systems produce similar results without significant differences, but small differences can be important depending on the requirements that are being pursued for a specific project. Analysing the results shown in Table 3 and Table 4, it can be seen how the precision, accuracy, and F-score values are occasionally better with the NDVI models, although the average values of accuracy, precision, recall, and F-score are better with the 12-band inputs. The best values in the comparison of one table with another are highlighted in bold. For a better understanding, graphs have been generated with the most important values to take into account: the F-score and loss (Figure 13a,b). Comparing the confusion matrices, the ideal situation would be to have the values of true positives and true negatives as high as possible. The best values, in general, are found in the results trained with the 12 bands.

Analysing the values of accuracy, precision, recall, and F-score, shown in Table 3, as well as the time values, shown in Table 2, we see that, proportionally, there is no significant difference in the models of the ten crops. Different results are obtained due to the number of examples used to train the models and the size of the evaluation period (see Table 1). Taking this second factor into account, we can group the ten crops into permanent (rapeseed, chickpea, alfalfa, olive, vineyard, and vineyard-olive) and nonpermanent (corn, rice, tomato, and onion/garlic/leek). Taking this into account, we can observe that permanent crops have a mean F-score of 0.9004 with the NDVI model, while the 12-band model gives a mean result of 0.9173, the latter being 1.21% better. As for the nonpermanent crops, we found an F-score of 0.9325 with the NDVI model and 0.9304 with the 12-band model, with the NDVI model being 0.19% better.

Overall, the 12-band models give more consistent results, but the differences, on many occasions, are marginal: the F-score is only 1% better for 12-band models compared to models trained with the NDVI. Additionally, the loss is better in NDVI models by 1%. The main distinction between the two is the amount of time spent on training (Figure 13c), where the training time is shorter in the models generated for the NDVI.

According to the training times shown and calculated in Table 2, it can be interpreted that the training time of the NDVI models is between 37.5% and 87.09% faster than that of the 12-band model, resulting in an average value of 59.35%. These values show that the NDVI models are quicker to train than the 12-band models. These times may not be considered important when training one or two models, but training hundreds of models can lead to significant gains in terms of time, amounting to hours or even days.

A diagram (Figure 14) shows the F-score achieved by the time it took to train each crop model, both 12-band and the NDVI. Although good results are obtained in both cases for all crops, it can be seen that, overall, the correlation between the training time and F-score is higher in those models that use the 12 bands.

## 4. Conclusions

This study aimed at finding the advantages and disadvantages of the use of the 12 bands in the Sentinel or the individual use of the NDVI for the detection of crops through satellite images. As we mentioned in the introduction, no similar studies have been found that directly compare the efficiency (regarding various factors) of using data of the 12 bands vs. the NDVI. The results obtained in the experiments with the ten types of crops have consistent values. We hope that this work will serve as a basis for other future experiments in this area. The main contributions of this work are as follows: (1) the results of a comparative study of crop identification using data from the 12 Sentinel-2 bands vs. the NDVI; (2) a method that can serve as the basis for other comparative studies that want to perform similar experiments; and (3) an analysis of some key factors to consider when undertaking a massive crop identification project.

For this study, we selected the necessary dates, downloaded Sentinel-2 images from 1 October 2020 to 15 July 2021, clipping them to reduce their size (5 km × 5 km), and extracted the bands for the analysis and calculation of the NDVI. Once the data had been obtained, they were divided into training, validation, and testing for the inputs of the different neural networks. Multiple experiments were carried out, reaching the best results for each of the experiments: 128–64–32 for the models whose input was the 12 bands and an architecture of 32–16–8 for the NDVI model. The results obtained for the data of the 12 bands are summarised as an average of loss of 0.2630, accuracy of 0.9334, precision of 0.9471, recall of 0.9184, and F-score of 0.9319. The average of the values of the results obtained with the NDVI are a loss of 0.2474, accuracy of 0.9141, precision of 0.9152, recall of 0.9133, and F-score of 0.9136.

It has been shown that both models can produce reliable results for pixel-by-pixel crop detection. The main difference is that, for the NDVI models, there had to be a preprocessing of the data beforehand to calculate this indicator, as it is an indicator that does not come by default in the original Sentinel-2 images. The 12-band models need more time to train because entry of the data of the twelve bands is necessary, but they are also more accurate in their results. If maximum accuracy is needed, it is recommended to use the 12-band model, but if only the NDVI is available, equivalent results can be achieved, which are less accurate but quicker to obtain.

To carry out this modelling study, a large amount of data referring to the crops of thousands of available plots offered by the administration of Extremadura (Spain) was available. Additionally, most of the images used to generate the models were useful, due to the low presence of clouds in this region (as described in Section 2.1). Thus, with many data as input sources, the generated models have high efficiency values. Thereby, the study counted on a considerable number of examples to generate the models, using multiple images in proper time periods and models that assess the coincidence of a declared crop with its corresponding model. Although the results cannot be generalised to all situations and data volumes, the study can be considered by system designers and researchers to evaluate the advantages and disadvantages of choosing one of the two approaches.

Analysing the data from the experiments, it can be concluded that using the 12 bands instead of only the NDVI provides better accuracy in the identification of crops, but to a minimal degree. It would only be advisable when seeking maximum precision levels. Regarding the calculation times, it will depend on the hardware resources. On the one hand, it is faster to generate neural network models using the NDVI, since the inputs are simpler; however, on the other hand, the calculation of the index itself entails computation time. Regarding the optimisation of storage, it would clearly be advisable to use the NDVI index against the storage of the 12 bands of Sentinel-2.

## Figures and Tables

**Figure 1 sensors-23-07132-f001:**
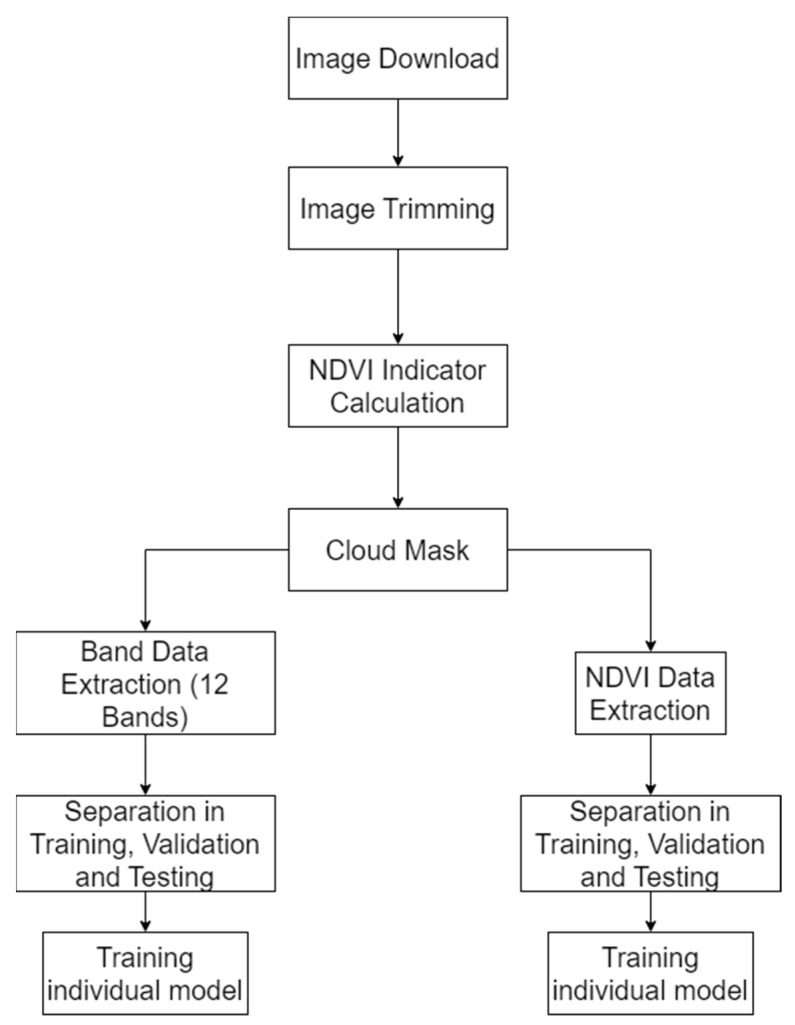
Overview of the main processes of the method for the double experiment.

**Figure 2 sensors-23-07132-f002:**
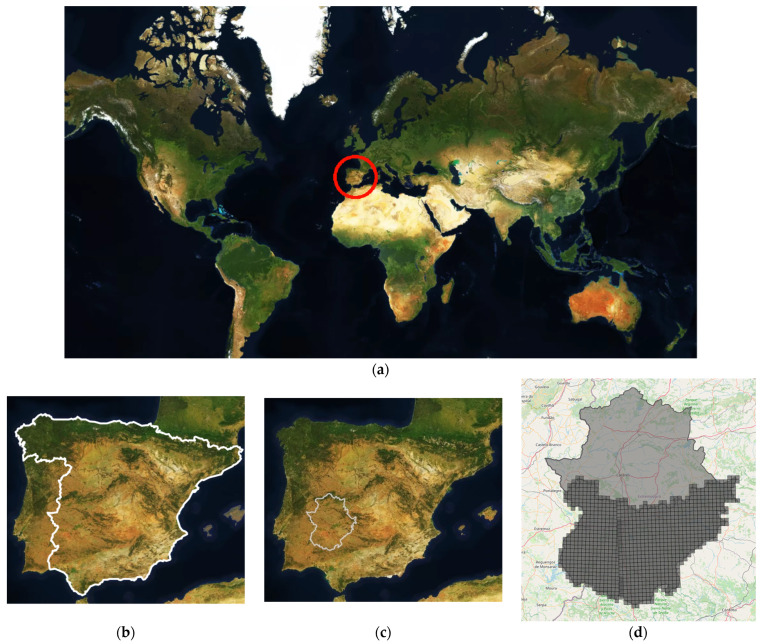
(**a**) World map with the Iberian Peninsula marked with a red circle (40°12′30.6″ N, 3°42′46.8″ W). (**b**) The Iberian Peninsula. Spain is outlined in white. (**c**) The Extremadura region is outlined in white (39°12′49.8″ N, 6°05′40.7″ W). (**d**) The light grey area is the entire Extremadura region, and the dark grey area is the work area (Badajoz region, Spain), divided into zones.

**Figure 3 sensors-23-07132-f003:**
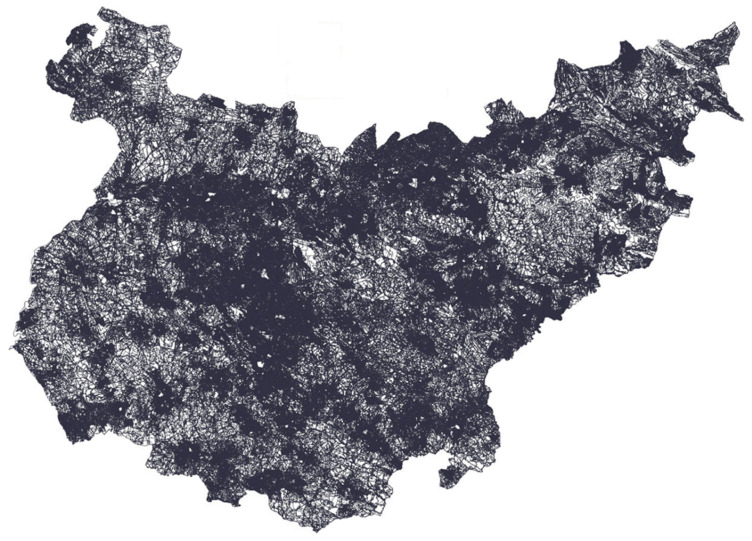
Geographical representation of the geolocation of the study plots (549,579 plots). Region of Badajoz (38°40′0″ N, 6°10′0″ W), Spain. The whole image is formed with the same grey colour lines representing the geometric feature of every studied plot. Where the lines seem to be darker, it is due to a greater density of plots in the area.

**Figure 4 sensors-23-07132-f004:**
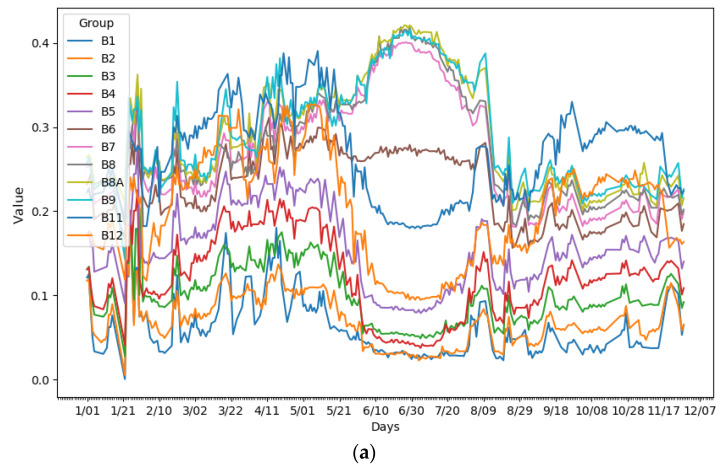
(**a**) Representation of the 12 bands of the corn crop during 2021. (**b**) Representation of the NDVI of the corn crop during 2021.

**Figure 5 sensors-23-07132-f005:**
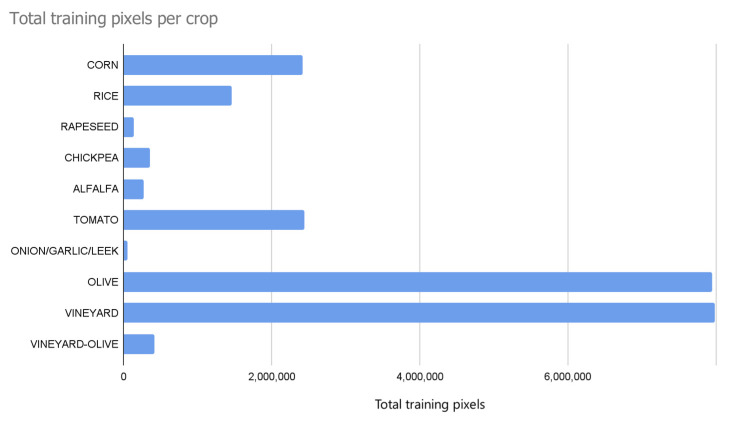
Representation of numbers of training pixels for each crop in the experiments.

**Figure 6 sensors-23-07132-f006:**
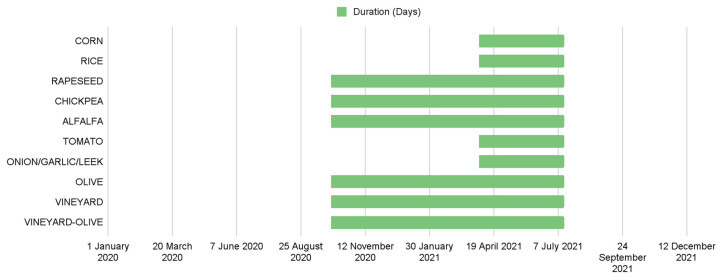
Graphic representation of the duration of the periods.

**Figure 7 sensors-23-07132-f007:**
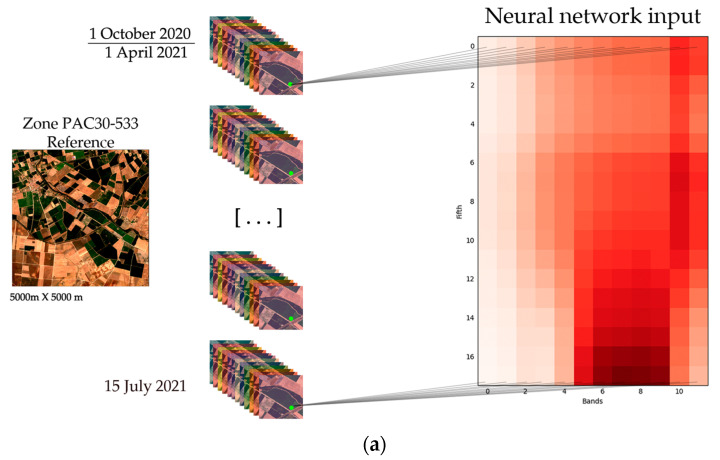
(**a**) Representation of the extraction of the 12 bands and temporal information for each of the neural network inputs. The colour gradient indicates the band values (normalised from 0, represented as white, to 1, represented as red). (**b**) Representation of the extraction of the NDVI and temporal information for each of the neural network inputs. The colour gradient indicates the band values (normalised from 0, represented as white, to 1, represented as red).

**Figure 8 sensors-23-07132-f008:**
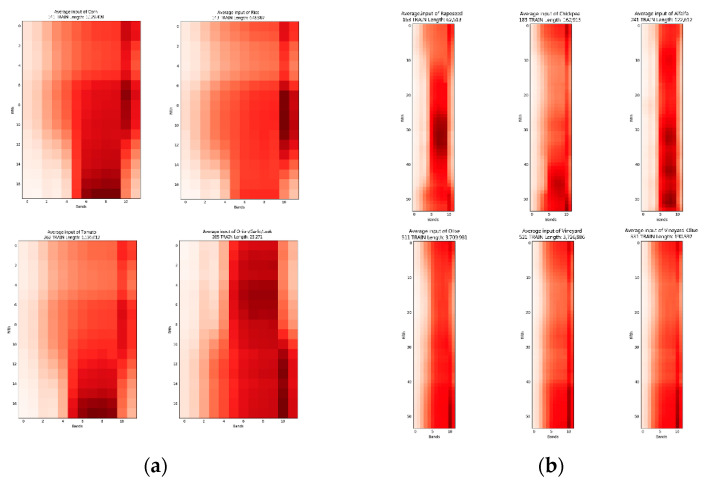
Representation of the average neural network inputs for each crop using the 12 bands. (**a**) Periods from 1 October 2020 to 15 July 2021. (**b**) Periods from 1 April 2021 to 15 July 2021.

**Figure 9 sensors-23-07132-f009:**
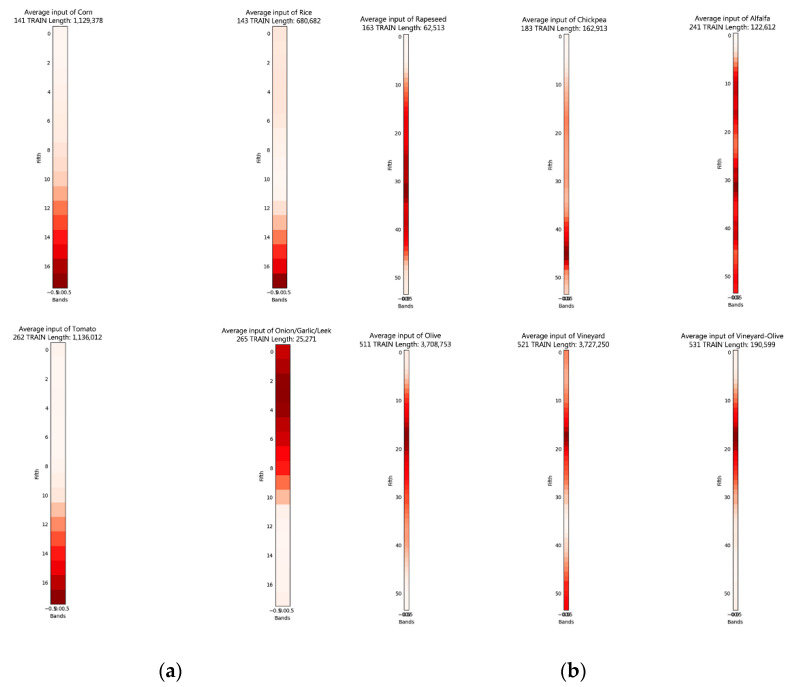
Representation of the average neural network inputs for each crop using the NDVI. (**a**) Periods from 1 October 2020 to 15 July 2021. (**b**) Periods from 1 April 2021 to 15 July 2021.

**Figure 10 sensors-23-07132-f010:**
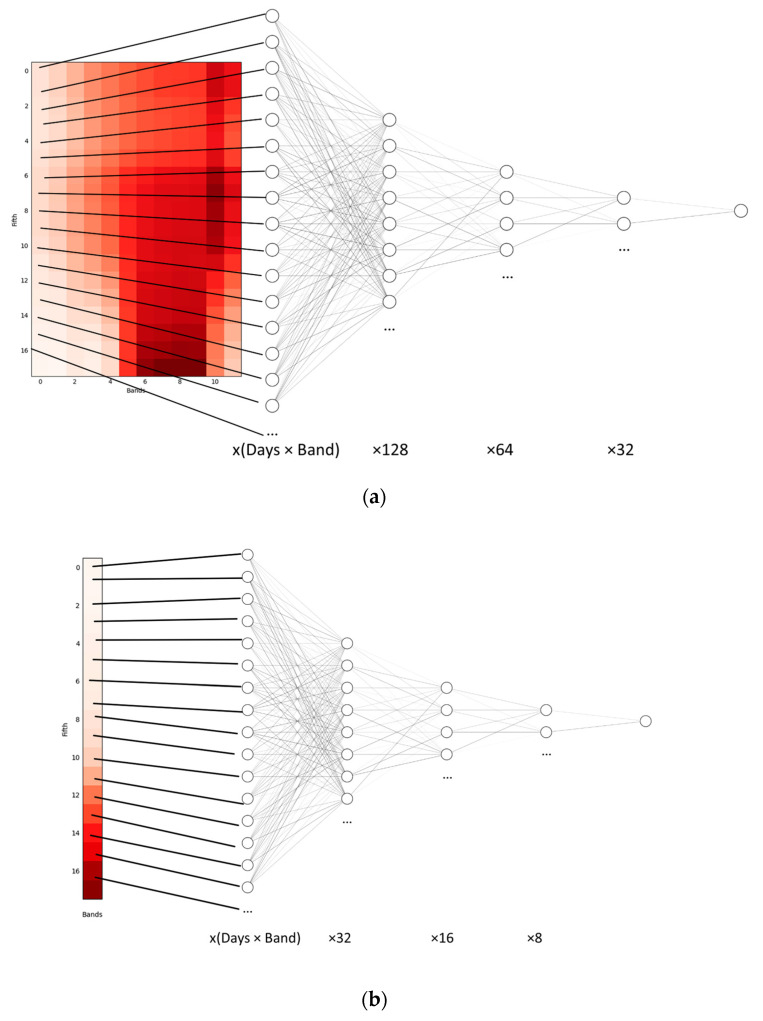
(**a**) Twelve-band model architecture (128–64–32). (**b**) NDVI model architecture (32–16–8).

**Figure 11 sensors-23-07132-f011:**
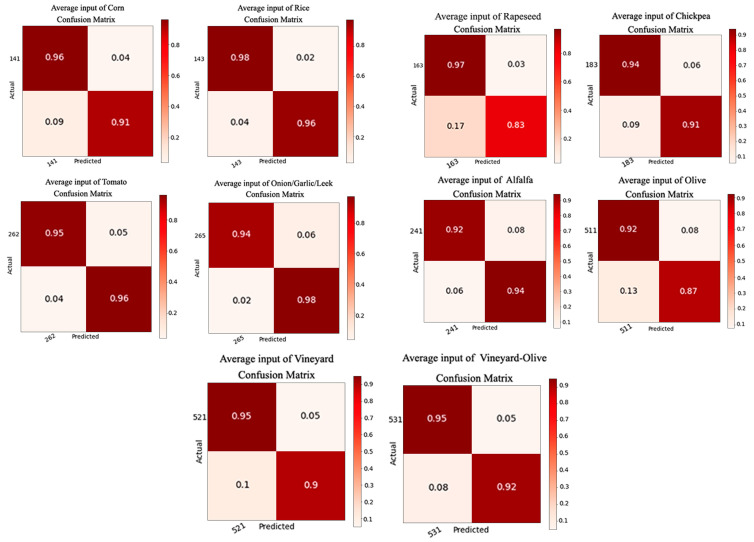
Confusion matrix generated for the results of the 12-band models.

**Figure 12 sensors-23-07132-f012:**
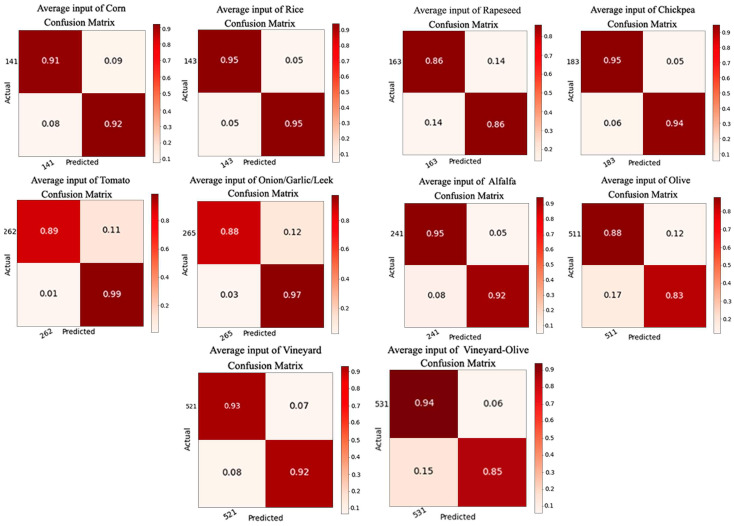
Confusion matrix generated for the results of the NDVI models.

**Figure 13 sensors-23-07132-f013:**
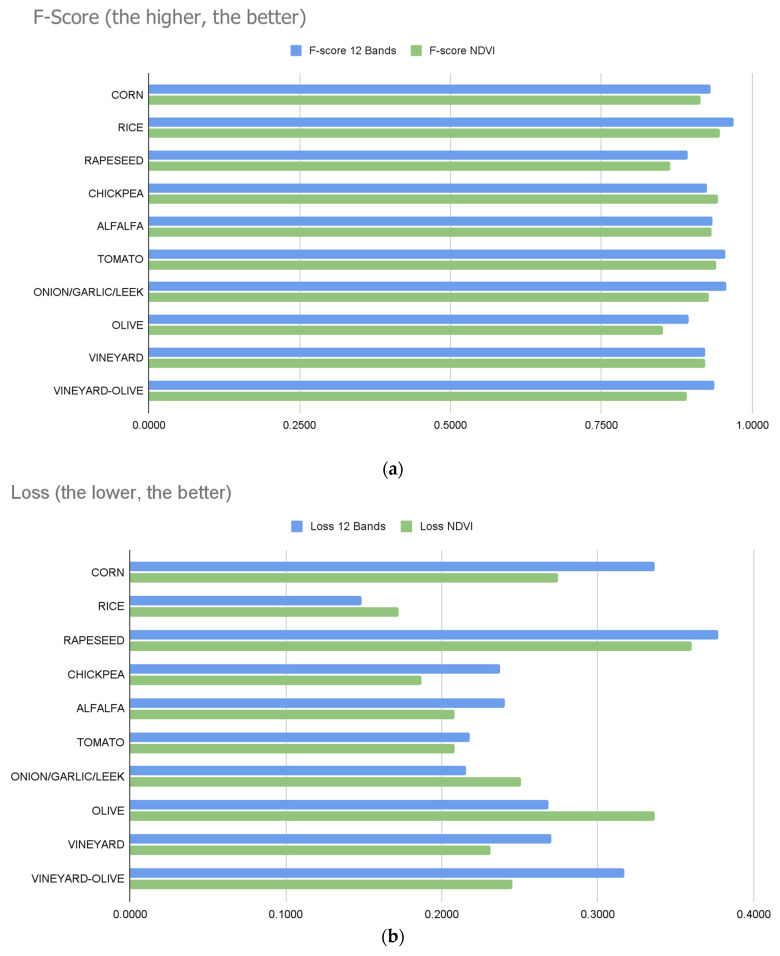
(**a**) Comparison of the F-scale measurement between the NDVI and 12-band models. (**b**) Comparison of the loss measurement between the NDVI and 12-band models. (**c**) Comparison of the training time between the NDVI and 12-band models.

**Figure 14 sensors-23-07132-f014:**
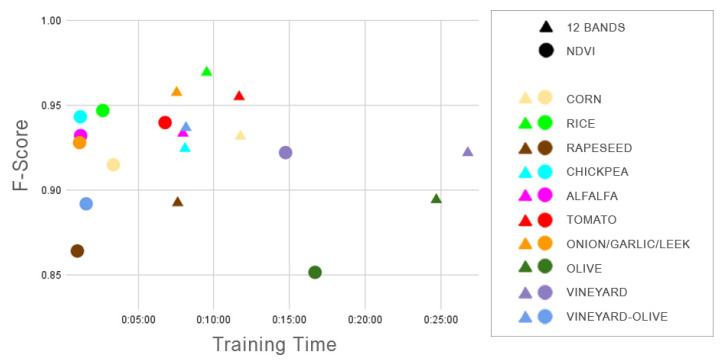
The F-score obtained by each crop following their training times is represented graphically.

**Table 1 sensors-23-07132-t001:** Start date and end date of the periods saved in the database and the number of pixels corresponding to each stage of the model.

Crop Name	Period	Start Date	End Date	Training Pixels	Validation Pixels	Testing Pixels
Corn	2	1 April 2021	15 July 2021	2,420,130	484,026	484,026
Rice	2	1 April 2021	15 July 2021	1,454,960	290,992	290,992
Rapeseed	1	1 October 2020	15 July 2021	133,940	26,788	26,788
Chickpea	1	1 October 2020	15 July 2021	349,080	69,816	69,816
Alfalfa	1	1 October 2020	15 July 2021	262,730	52,546	52,546
Tomato	2	1 April 2021	15 July 2021	2,434,290	486,858	486,858
Onion/garlic/leek	2	1 April 2021	15 July 2021	54,110	10,822	10,822
Olive	1	1 October 2020	15 July 2021	7,949,940	1,589,988	1,589,988
Vineyard	1	1 October 2020	15 July 2021	7,986,370	1,597,274	1,597,274
Vineyard-olive	1	1 October 2020	15 July 2021	408,400	81,680	81,680
Total values	-	-	-	23,453,950	4,690,790	4,690,790

**Table 2 sensors-23-07132-t002:** Training time for each of the crops for the 12-band and NDVI models. The best values are highlighted in bold.

Crop	Training Time 12-Band Model	Training Time NDVI Model
Corn	11 min 46 s	**03 min 22 s**
Rice	09 min 22 s	**02 min 40 s**
Rapeseed	07 min 37 s	**00 min 59 s**
Chickpea	08 min 06 s	**01 min 11 s**
Alfalfa	07 min 58 s	**01 min 12 s**
Tomato	11 min 41 s	**06 min 47 s**
Onion/garlic/leek	07 min 33 s	**01 min 08 s**
Olive	24 min 43 s	**16 min 42 s**
Vineyard	26 min 48 s	**14 min 45 s**
Vineyard-olive	08 min 10 s	**01 min 34 s**
**Average values**	12 min 23 s	**05 min 02 s**

**Table 3 sensors-23-07132-t003:** Training results for the models of the 12 bands. The best values in the comparison of Table 3 (12 bands) versus Table 4 (NDVI index) are highlighted in bold.

Crop Name	Period	Training Positive Pixels	Loss	Overall Accuracy	Precision	Recall	F-Score
Corn	2	1,210,065	0.3363	**0.9336**	**0.9563**	0.9086	**0.9319**
Rice	2	727,480	**0.1484**	**0.9698**	**0.9764**	**0.9630**	**0.9696**
Rapeseed	1	66,970	0.3775	**0.9005**	**0.9685**	0.8280	**0.8927**
Chickpea	1	174,540	0.2376	**0.9258**	0.9386	0.9112	0.9247
Alfalfa	1	131,365	0.2406	0.9330	0.9243	**0.9433**	**0.9337**
Tomato	2	1,217,145	0.2181	**0.9549**	**0.9503**	0.9601	**0.9552**
Onion/garlic/leek	2	27,055	**0.2153**	**0.9568**	**0.9389**	0.9773	**0.9577**
Olive	1	3,974,970	**0.2687**	**0.8970**	**0.9170**	**0.8731**	**0.8945**
Vineyard	1	3,993,185	0.2705	**0.9242**	**0.9481**	0.8976	0.9222
Vineyard-olive	1	204,200	0.3170	**0.9381**	**0.9528**	**0.9218**	**0.9370**
**Average values**	-	1,172,698	0.2630	**0.9334**	**0.9471**	**0.9184**	**0.9319**

**Table 4 sensors-23-07132-t004:** Training results for the models of the NDVI. The best values in the comparison of Table 3 (12 bands) versus Table 4 (NDVI index) are highlighted in bold.

Crop Name	Period	Training Positive Pixels	Loss	Overall Accuracy	Precision	Recall	F-Score
Corn	2	1,210,030	0.2743	0.9149	0.9136	**0.9165**	0.9151
Rice	2	729,295	0.1723	0.9471	0.9484	0.9457	0.9471
Rapeseed	1	66,970	**0.3600**	0.8642	0.8639	**0.8646**	0.8643
Chickpea	1	174,540	**0.1868**	0.9437	**0.9491**	**0.9378**	**0.9434**
Alfalfa	1	131,365	**0.2082**	**0.9335**	**0.9499**	0.9153	0.9323
Tomato	2	1,217,145	**0.2083**	0.9371	0.8984	**0.9856**	0.9400
Onion/garlic/leek	2	27,055	0.2510	0.9252	0.8927	**0.9667**	0.9282
Olive	1	3,973,655	0.3367	0.8555	0.8746	0.8298	0.8517
Vineyard	1	3,993,470	**0.2311**	0.9227	0.9271	**0.9175**	**0.9223**
Vineyard-olive	1	204,205	**0.2451**	0.8968	0.9348	0.8531	0.8921
Average values	-	1,172,773	0.2474	0.9141	0.9152	0.9133	0.9136

## Data Availability

The raw/processed data required to reproduce these findings cannot be shared at this time as the data also forms part of an ongoing study.

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
