# Peer review of "Evaluation of the Use of the 12 Bands vs. NDVI from Sentinel-2 Images for Crop Identification"

_sensors, 2023, doi:10.3390/s23167132_

Round 1

Reviewer 1 Report

Summary:

This paper analysed the neural network models trained with the Sentinel satellite’s 12 bands compared to only using the NDVI index, in order to choose the most suitable in terms of amount of storage, calculation time, accuracy and precision. This study may be of interest to administrations, businesses, land managers and researchers who use satellite images, especially if storage capacity and response times are limited.

General comments:

Despite, the authors have conducted a study well documented with appropriate citations, I recommend that the paper needs more revision. The main issue comes from the text and the research significance of the paper. The author must show and analyze the results comprehensively with the research problem in mind. The paper is simple and the innovation is insufficient.

In this work, it is a pity that the results of crop identification using multi-temporal and single-temporal data were not compared. Many studies have chosen to identify crops using time series data to improve accuracy, but it takes a long time to execute. I suggest that the authors could increase the comparison of identifying crops using different individual growth period data and whole growth period data. This has more research significance, including the need for early extraction of crop acreage and the widespread absence of remote sensing data.

Detailed comments:

1) Line 15: "This study may be of interest to administrations,", Line 18: "The results show that," ===> The summary should first introduce the results and then explain the practical significance.

2) Line 18-19: Please pay attention to grammar and typographical mistakes everywhere.

3) Introduction:

- The authors are invited to summarize the contributions of their work in a form of a list of short sentences.

- The authors are invited to add a short paragraph at the end of the introduction which describes the structure of the paper.  

- The introduction contains a large number of citations.

- The language is not accurate. For instance, Line 37: "to control the crops and improve production,", it is not the direct utility of crop identification.  

- The authors are invited to compare their contribution with similar existing works.  

4) Method:

- The accuracy of crop identification results is very important, the accuracy evaluation process must be introduced in a Method section.

- What happens to Sentinel data covered by clouds? A part on the data quality check is missing. You are mixing data presentation and preprocessing. I recommend that you add a section 2.3 preprocessing for example.

- Line 184-189: What are the spectral curve of maize measured in the field? Give references.

- Line 232-239: Could you add a comment on the impact of an interpolation of 5 days? Have you compared with other verified data sources like MODIS(MCD43A4)? Can you estimate the error of your interpolation? Give a reference for the interpolation method.

- The authors need to provide definitions of the different indices presented in Table 2.

5) Results and Discussion

- Line 318: Table 3. Training time for each of the crop for NDVI models.. Training time 12 Band model? It needs to be checked.

- In addition, the authors need to provide more explanation and discussion about the model results based on the crop categories they considered, now that different models for different crops has been established in their study.

- Line 361-362: In Table 3, there is no significant advantage in training time between 12 Band model and NDVI model. Why did the author come to the conclusion “The main distinction between the two is the amount of time spent on training (Figure 13c), where the training time is less in the models generated for the NDVI” .

Summary:

This paper analysed the neural network models trained with the Sentinel satellite’s 12 bands compared to only using the NDVI index, in order to choose the most suitable in terms of amount of storage, calculation time, accuracy and precision. This study may be of interest to administrations, businesses, land managers and researchers who use satellite images, especially if storage capacity and response times are limited.

General comments:

Despite, the authors have conducted a study well documented with appropriate citations, I recommend that the paper needs more revision. The main issue comes from the text and the research significance of the paper. The author must show and analyze the results comprehensively with the research problem in mind. The paper is simple and the innovation is insufficient.

In this work, it is a pity that the results of crop identification using multi-temporal and single-temporal data were not compared. Many studies have chosen to identify crops using time series data to improve accuracy, but it takes a long time to execute. I suggest that the authors could increase the comparison of identifying crops using different individual growth period data and whole growth period data. This has more research significance, including the need for early extraction of crop acreage and the widespread absence of remote sensing data.

Detailed comments:

1) Line 15: "This study may be of interest to administrations,", Line 18: "The results show that," ===> The summary should first introduce the results and then explain the practical significance.

2) Line 18-19: Please pay attention to grammar and typographical mistakes everywhere.

3) Introduction:

- The authors are invited to summarize the contributions of their work in a form of a list of short sentences.

- The authors are invited to add a short paragraph at the end of the introduction which describes the structure of the paper.  

- The introduction contains a large number of citations.

- The language is not accurate. For instance, Line 37: "to control the crops and improve production,", it is not the direct utility of crop identification.  

- The authors are invited to compare their contribution with similar existing works.  

4) Method:

- The accuracy of crop identification results is very important, the accuracy evaluation process must be introduced in a Method section.

- What happens to Sentinel data covered by clouds? A part on the data quality check is missing. You are mixing data presentation and preprocessing. I recommend that you add a section 2.3 preprocessing for example.

- Line 184-189: What are the spectral curve of maize measured in the field? Give references.

- Line 232-239: Could you add a comment on the impact of an interpolation of 5 days? Have you compared with other verified data sources like MODIS(MCD43A4)? Can you estimate the error of your interpolation? Give a reference for the interpolation method.

- The authors need to provide definitions of the different indices presented in Table 2.

5) Results and Discussion

- Line 318: Table 3. Training time for each of the crop for NDVI models.. Training time 12 Band model? It needs to be checked.

- In addition, the authors need to provide more explanation and discussion about the model results based on the crop categories they considered, now that different models for different crops has been established in their study.

- Line 361-362: In Table 3, there is no significant advantage in training time between 12 Band model and NDVI model. Why did the author come to the conclusion “The main distinction between the two is the amount of time spent on training (Figure 13c), where the training time is less in the models generated for the NDVI” .

Author Response

C1.1- General comments:

Despite, the authors have conducted a study well documented with appropriate citations, I recommend that the paper needs more revision. The main issue comes from the text and the research significance of the paper. The author must show and analyze the results comprehensively with the research problem in mind. The paper is simple and the innovation is insufficient.

In this work, it is a pity that the results of crop identification using multi-temporal and single-temporal data were not compared. Many studies have chosen to identify crops using time series data to improve accuracy, but it takes a long time to execute. I suggest that the authors could increase the comparison of identifying crops using different individual growth period data and whole growth period data. This has more research significance, including the need for early extraction of crop acreage and the widespread absence of remote sensing data.

A.1.1- We fully agree with the reviewer that it is essential to determine the appropriate analysis periods for crop identification processes and to be able to compare them. We have added in the article (3 paragraphs after the second paragraph of section 2) a brief explanation of one of the phases of the proposed method that is described in detail in the reference provided. In it, the comparison of the results of different time series of data that are used to determine the optimal period of identification of a crop can be analyzed.

C1.2- Line 15: "This study may be of interest to administrations,", Line 18: "The results show that," ===> The summary should first introduce the results and then explain the practical significance.

A.1.2- The summary has been reorganized.

C1.3- Line 18-19: Please pay attention to grammar and typographical mistakes everywhere.

A.1.3- All text has been reviewed by a native English speaker and some grammatical errors have been corrected.

C1.4- The authors are invited to summarize the contributions of their work in a form of a list of short sentences.

A.1.4 The main contributions of this work have been listed at the end of the first paragraph of Conclusions.

C1.5- The authors are invited to add a short paragraph at the end of the introduction which describes the structure of the paper.

A.1.5- The text has been added at the end of Section 1.

C1.6- The introduction contains a large number of citations.

A.1.6- The introduction has been reorganized in order to reduce the number of citations.

C1.7- The language is not accurate. For instance, Line 37: "to control the crops and improve production,", it is not the direct utility of crop identification.

A.1.7- It is true that it is not the direct use of crop detection, but additionally thanks to the use of it at pixel level, it is possible to notify the existence of other crops in the same enclosure or even to notify farmers of incidences in certain parts of the enclosure (such as incorrect crop growth). The text has been updated in the second paragraph of Section 1 to clarify this point.

C1.8- The authors are invited to compare their contribution with similar existing works.

A.1.8- In the Introduction we have collected references that are related to the identification of crops with the use of 12 bands and with the NDVI index. However, no papers have been found that directly compare the efficiency of both approaches. We think that our work can serve as a basis for future studies. We have clarified this point in the first paragraph of the conclusions.

C1.9- The accuracy of crop identification results is very important, the accuracy evaluation process must be introduced in a “Method” section.

A.1.9- We are able to evaluate the performance of the different metrics using the ground truth data provided by the administration inspectors. We have explained this process at the end of Section 2.

C1.10- What happens to Sentinel data covered by clouds? A part on the data quality check is missing. You are mixing data presentation and preprocessing. I recommend that you add a section “2.3 preprocessing” for example

A.1.10- Sentinel data covered by clouds are discarded using the scene classification mask provided by the Sentinel images. In addition, a Hampel Filter is applied as data quality check, detecting remaining anomalous data (outliers) and replacing them. This was explained in the first citation of second paragraph (Section 2.6), but a clarification has been added in the same paragraph..

Additionally, a new section has been added to separate data presentation and preprocessing.

C1.11- Line 184-189: What are the spectral curve of maize measured in the field? Give references.

A.1.11- Figures 4a and 4b have been generated by us, averaging the reflectance values of each pixel of the images over the time period of the maize crop during 2021 at the different date. A clarification has been added before the Figures 4a and 4b.

C1.12- Line 232-239: Could you add a comment on the impact of an interpolation of 5 days? Have you compared with other verified data sources like MODIS(MCD43A4)? Can you estimate the error of your interpolation? Give a reference for the interpolation method.

A.1.12- It is not very common that an interpolation is needed in the work area, so the impact is not relevant.This is also the reason why no other data sources have been used. Different clarifications have been added to the first and third paragraph of Section 2.6.

C1.13- The authors need to provide definitions of the different indices presented in Table 2.

A.1.13- Definitions have been added at the end of Section 2.7.

C1.14- Line 318: “Table 3. Training time for each of the crop for NDVI models.“. Training time 12 Band model? It needs to be checked.

A.1.14- The text has been updated in Table 3.

C1.15- In addition, the authors need to provide more explanation and discussion about the model results based on the crop categories they considered, now that different models for different crops has been established in their study.

A.1.15- Paragraph 5 has been added in Section 3.2 on the results obtained taking into account the models obtained for the different crops.

C1.16- Line 361-362: In Table 3, there is no significant advantage in training time between 12 Band model and NDVI model. Why did the author come to the conclusion “The main distinction between the two is the amount of time spent on training (Figure 13c), where the training time is less in the models generated for the NDVI” .

A.1.16- Interpreting the values in Table 3, where the training times are shown NDVI model is 59.35% faster than the one trained with the 12 bands. If only one model is trained, this may not be significant, but when training hundreds of models, the gain can be hours or even days. The text has been added in the seventh paragraph of the Section 3.2.

Reviewer 2 Report

Good job! Please check my comments below:

1.      Can you provide results value in abstract? What do you mean “little significant”?

2.      Did you mean reflectance of the 12 bands? You may need to mention it in the introduction.

3.      Line 48, include the full name of NDVI at first appearance.

4.      Why did you choose NDVI instead of other VIs?

5.      Please add the latitude and longitude of geographic maps.

6.      Lines 168-169, you normalized the original date to 0-1, what is the original data range?

7.      How do you deal with the zones without the ten crops you listed?

8.      Why there is a NDVI small peak of figure 4b around 1/21?

Author Response

C.2.1 - Can you provide results value in abstract? What do you mean “little significant”?

A.2.1 - The results have been added to the abstract. Additionally, the abstract has been rewritten in order to clarify its content.

C.2.2 - Did you mean reflectance of the 12 bands? You may need to mention it in the introduction.

A.2.2 - The information has been added in the third paragraph of the introduction.

C.2.3 - Line 48, include the full name of NDVI at first appearance.

A.2.3 - A modification has been made in the third paragraph of the introduction

C.2.4 - Why did you choose NDVI instead of other VIs?

A.2.4 - The NDVI index is one of the most used in remote sensing, as it is collected in the references that appear in the Introduction, so we understand that this work can be more useful. A clarification has been added in the third paragraph of the introduction.

C.2.5 - Please add the latitude and longitude of geographic maps.

A.2.5 - Coordinates have been added in figure 2.

C.2.6 - Lines 168-169, you normalized the original date to 0-1, what is the original data range?

A.2.6 - According to the ESA documentation (https://step.esa.int/thirdparties/sen2cor/2.5.5/docs/S2-PDGS-MPC-L2A-IODD-V2.5.5.pdf) , reflectance values are between 0 and 1, but a manual study of the original images and with the support of Sentinel Hub documentation (https://docs.sentinel-hub.com/api/latest/data/sentinel-2-l2a/) , showed that reflectance values can be above 1. This explanation has been added in the first paragraph of Section 2.3

C.2.7 - How do you deal with the zones without the ten crops you listed?

A.2.7 - In this study we work at the pixel level, therefore, pixels that do not belong to the ten crops listed are ignored. A clarification has been added in the second paragraph in Section 2.5, before Table 1.

C.2.8 - Why there is a NDVI small peak of figure 4b around 1/21?

A.2.8 - These peaks are caused by anomalous data due to atmospheric phenomena, such as clouds. An explanation has been added in the third paragraph of the Section 2.3.

Round 2

Reviewer 1 Report

This paper has made targeted revisions to the previous comments.

This paper has made targeted revisions to the previous comments.

Reviewer 2 Report

Great work!